# A Systematic Review of Non-Pharmacological Interventions to Improve Gait Asymmetries in Neurological Populations

**Krista G. Meder** [1,*] , **Chanel T. LoJacono** [2] **and Christopher K. Rhea** [1]

1   Virtual Environment for Assessment and Rehabilitation (VEAR) Lab, Department of Kinesiology,
    University of North Carolina at Greensboro, Greensboro, NC 27412, USA; ckrhea@uncg.edu
2   Department of Kinesiology, Missouri Southern State University, Joplin, MO 64801, USA; lojacono-c@mssu.edu
*   Correspondence: kgmeder@uncg.edu

**Abstract:** Gait asymmetries are commonly observed in neurological populations and linked to decreased gait velocity, balance decrements, increased fall risk, and heightened metabolic cost. Interventions designed to improve gait asymmetries have varying methods and results. The purpose of this systematic review was to investigate non-pharmacological interventions to improve gait asymmetries in neurological populations. Keyword searches were conducted using PubMed, CINAHL, and Academic Search Complete. The search yielded 14 studies for inclusion. Gait was assessed using 3D motion capture systems ($n = 7$), pressure-sensitive mats (e.g., GAITRite; $n = 5$), and positional sensors ($n = 2$). The gait variables most commonly analyzed for asymmetry were step length ($n = 11$), stance time ($n = 9$), and swing time ($n = 5$). Interventions to improve gait asymmetries predominantly used gait training techniques via a split-belt treadmill ($n = 6$), followed by insoles/orthoses ($n = 3$). The literature suggests that a wide range of methods can be used to improve spatiotemporal asymmetries. However, future research should further examine kinematic and kinetic gait asymmetries. Additionally, researchers should explore the necessary frequency and duration of various intervention strategies to achieve the greatest improvement in gait asymmetries, and to determine the best symmetry equation for quantifying gait asymmetries.

**Keywords:** gait; locomotion; asymmetry; stroke; Parkinson's disease

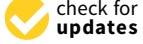



## 1. Introduction

### 1.1. Rationale

Musculoskeletal and neurological disorders such as osteoarthritis, amputations, Parkinson's disease (PD), and stroke may cause decrements in walking and balance, leading to the development of gait asymmetries. Gait asymmetry is defined as differences in the bilateral behavior of the lower extremities during walking [1], and is an important measure for assessing gait quality [2]. Increases in gait asymmetries are linked to decreased gait velocity [2,3], decreased balance [4], decreased bone mineral density in the paretic limb [5], increased metabolic cost [6–8], and elevated fall risk [3,4,9]—leading to a decreased ability to complete activities of daily living [10] and an overall decrease in quality of life [11]. Thus, a major goal of gait rehabilitation in clinical populations is to reduce gait asymmetries [2,12]. However, interventions vary on what methods work to specifically improve gait asymmetries.

Pharmacological interventions have been shown to alter gait kinematics such as gait speed, gait variability, range of motion, and step length [12,13]. Individuals with PD can experience freezing of gait (FOG), for which dopaminergic treatments are used. For example, levodopa—a dopamine replacement therapy—has been shown to be effective in improving scores on the Unified Parkinson's Disease Rating Scale (UPDRS), including reduced FOG, and in improving stride time and variability [13]. Roemmich and colleagues examined split-belt treadmill training in conjunction with dopaminergic treatment and

showed a greater magnitude of adaptation (i.e., larger response to the perturbation) with the medication on step length and hip and knee range of motion [12]. Additionally, gait speed increased when using dopaminergic treatment [12]. These findings are not isolated to levodopa, as there are other pharmacological approaches to addressing gait challenges—particularly in individuals with PD [14]. However, gait symmetry is typically not a primary outcome variable in these studies and no pharmacological interventions are known to enhance gait symmetry [15–17].

Many non-pharmacological interventions have been used to enhance gait function and symmetry [18], including deep brain stimulation [19]. However, these interventions can sometimes be invasive, as is the case with deep brain stimulation. Thus, non-invasive and non-pharmacological interventions are desirable for many people. One such example is split-belt treadmill training, which consists of two belts on a treadmill that operate independently for each limb. The belts can be programmed to move at the same speed (which simulates a typical treadmill) or at separate speeds, which can evoke gait asymmetries, and in some cases, is used to enhance symmetry in those who have adopted an asymmetrical gait pattern. For example, split-belt treadmill training has been used for rehabilitation in individuals post-stroke with some success. Split-belt treadmill training has been shown to temporarily induce symmetry [20] and transfer the improved symmetry to overground walking [21]. While the mechanism of pathology is different in stroke and PD, researchers have used similar protocols for gait improvements. Roemmich and colleagues investigated the use of a split-belt treadmill in people with Parkinson's disease and utilized the same protocol as in the post-stroke studies, which resulted in similar gait asymmetry improvements [22]. In addition to split-belt treadmill training, body weight support training has been employed with stroke survivors. Body weight support training using verbal and manual assistance has shown improved bilateral coordination of the shank and foot, but did not show reductions in gait asymmetries [23]. Moreover, assistive devices have shown to improve body weight on the paretic limb and improve gait speed [24,25]. For example, Chen and colleagues have shown that insoles improve stance asymmetry, but not gait asymmetry [24]. In contrast, several other studies have shown improvements in gait asymmetries with the use of insoles or orthoses [25,26]. Due to the various techniques for altering gait asymmetry, our aim was to investigate the current literature for the most common and successful methods for improving gait asymmetries in neurological populations.

### *1.2. Objective*

The purpose of this systematic review was to investigate non-pharmacological interventions to improve gait asymmetries.

### **2. Methods**
### *2.1. Systematic Review Protocol*

This systematic review was performed according to the Preferred Reporting Items for Systematic Reviews and Meta-Analyses (PRISMA) guidelines [27].

### *2.2. Inclusion/Exclusion Criteria*

The inclusion criteria involved full-text and English language articles that focused on the following: (1) adults, (2) lower extremity motion, (3) asymmetrical gait, and (4) direct intervention to correct gait asymmetries. Studies were excluded for the following reasons: (1) non-clinical diagnoses of gait asymmetry, (2) amputees, (3) surgery on the lower extremity, (4) non-human populations, (5) no direct intervention for improving gait asymmetries, (6) child or adolescent populations, and (7) upper extremity motion. Studies with amputee and post-surgical populations were excluded from this review because of the additional factors that could confound interventions to improve gait asymmetry.

### 2.3. Search Strategy

A keyword specific search of PubMed, CINAHL, and Academic Search Complete databases was conducted on 21 June 2021 by KM and CL for articles in English with full texts. The search index was Title and Abstract for CINAHL and Academic Search Complete, and Title and Other Term in PubMed. The specific search algorithm is provided in Table 1. Research articles from January 2009 to December 2020 were obtained. The articles were managed using Microsoft Excel (Microsoft, Redmond, WA, USA) and Zotero (https://www.zotero.org; Corporation for Digital Scholarship, Vienna, VA, USA) last accessed on 21 December 2021.

**Table 1.** Search algorithm for each database.

| Database | Search Index | Search Terms |
|---|---|---|
| PubMed | Title and Other Term | (gait or ambulation or walking or mobility or locomotor or locomotion) AND (symmetry or asymmetry or symmetries or asymmetries or symmetrical or asymmetrical) |
| CINAHL | Title and Abstract | (gait or ambulation or walking or mobility or locomotor or locomotion) AND (symmetry or asymmetry or symmetries or asymmetries or symmetrical or asymmetrical) |
| Academic Search Complete | Title and Abstract | (gait or ambulation or walking or mobility or locomotor or locomotion) AND (symmetry or asymmetry or symmetries or asymmetries or symmetrical or asymmetrical) |

### 2.4. Study Selection

Article duplicates were removed. An initial screening of the studies via title and abstract based on the inclusion and exclusion criteria was performed. Studies not eliminated were further reviewed by full text for the inclusion and exclusion criteria. The titles, abstracts, and full text articles were assessed independently for eligibility by two reviewers, KM and CL. Then, the two reviewers cross-referenced eligible articles. In the case of disagreement, a discussion between the two primary reviewers either came to an agreement or was resolved by the third author (CK).

### 2.5. Data Extraction

The elements extracted from each included article were population type, sample size, age, gait variables, gait intervention, gait assessment, gait symmetry equation, and results.

## 3. Results

### 3.1. Study Search Selection

Figure 1 illustrates the article search and screening process. A total of 200 articles were found. After duplicates were removed, 161 articles were screened via their title and abstract based on the inclusion/exclusion criteria—resulting in 17 studies assessed by full text. Three articles did not meet the criteria because the studies either did not have a direct intervention to improve gait asymmetries or did not have a clinical diagnosis of gait asymmetry. As a result, 14 articles were included in the review [28–41].

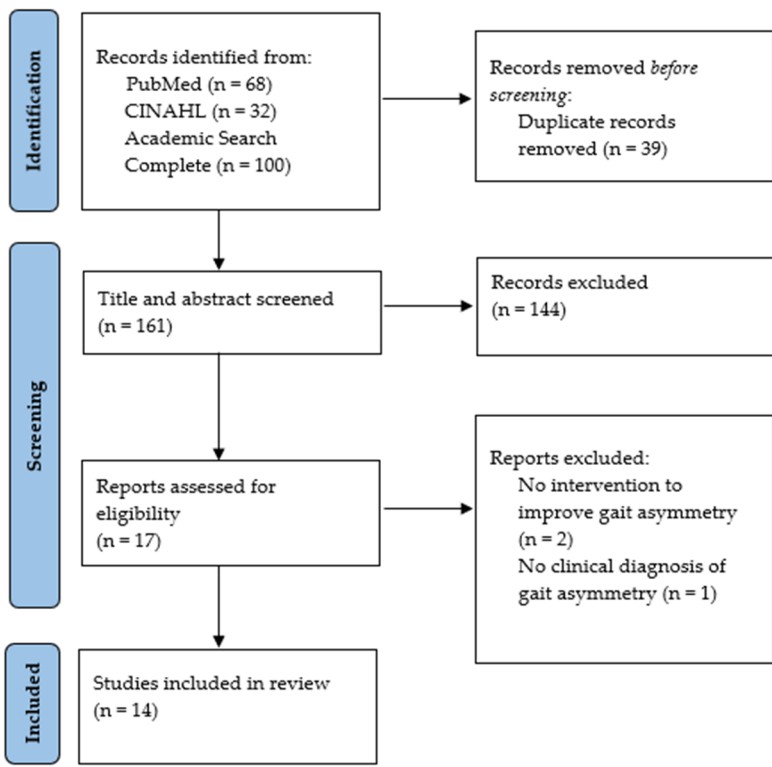

**Figure 1.** PRISMA flow diagram.

*3.2. Demographic Information*

All studies examined populations with clinical diagnoses of gait asymmetries, as shown in Table 2. The populations examined included stroke (*n* = 10), Parkinson's disease (*n* = 3), and poliomyelitis (*n* = 1). The majority of sample sizes were 20 participants or less (*n* = 12). Two studies contained a wide range of ages, but ages in the late 40s to early 70s were the most reported (*n* = 11). Brodie and colleagues [31] was the only study to have two control groups: healthy age-matched and healthy young controls. One study reported only the ages of the healthy control participants and not of the stroke population [28].

**Table 2.** Summary of included articles investigating interventions to improve gait asymmetries in clinical populations.

| References | N (Clinical/Control) | Population | Age (Years) |
|---|---|---|---|
| Afzal et al., 2015 | 4/5 | Stroke | NR/26.2 ± 3.27 |
| Arazapour et al., 2016 | 7 | Poliomyelitis | 47–57 |
| Beauchamp et al., 2009 | 14 | Stroke | 61 ± 10 |
| Brodie et al., 2015 | 10/11, 9 | Parkinson's Disease | 67/66, 30 |
| Fasano et al., 2016 | 20 | Parkinson's Disease | 60.5 ± 8.8 |
| Kahn et al., 2009 | 18 | Stroke | 34–74 |
| Lewek et al., 2012 | 2 | Stroke | 53, 60 |
| Lewek et al., 2018 | 48 | Stroke | 59 ± 12 |
| Little et al., 2020 | 39 | Stroke | 61.3 ± 11.4 |
| Ma et al., 2018 | 17 | Stroke | 56.2 ± 7.3 |
| Mieville et al., 2018 | 20 | Stroke | 49.4 ± 13.2 |
| Padmanabhan et al., 2020 | 9 | Stroke | 54 ± 4 |
| Yen et al., 2015 | 10 | Stroke | 36–67 |
| Zanardi et al., 2019 | 14 | Parkinson's Disease | 66.8 ± 9.6 |

NR = Not Reported.

### 3.3. Interventions and Assessments of Gait Asymmetry

Gait was assessed using 3D motion capture systems ($n = 7$), pressure-sensitive mats (e.g., GAITRite; $n = 5$), and positional sensors ($n = 2$). A depiction of the assessments and interventions for each study can be found in Table 3. The gait variables most commonly analyzed for asymmetry were step length (SL; $n = 11$), stance time (ST; $n = 9$), and swing time (SWT; $n = 5$). Four studies measured single or double limb support time/phase and two of these studies examined speed. Three studies measured joint kinematics.

**Table 3.** Summary of included articles investigating interventions to improve gait asymmetries in clinical populations.

| References | Gait Variables | Gait Assessment | Intervention |
|---|---|---|---|
| Afzal et al., 2015 | ST | Force-sensitive resistors | Insoles with vibrotactors to provide feedback. Healthy group walked 10 m in seven scenarios. Stroke group walked 6 m in three scenarios. Both groups performed two trials per scenario. |
| Arazapour et al., 2016 | SW, SL, ST, SWT, double-limb support time, stance phase, walking speed, knee flexion | 3D Motion Capture (VICON) | Compared drop-locked orthosis to new powered Knee Ankle Foot Orthosis (KAFO). Three trials of 6 m walking at preferred speed for each orthosis. |
| Beauchamp et al., 2009 | ST, SWT | Pressure-Sensitive Mat (GAITRite) | Walked 6 m in all three conditions (1) without cane, (2) single-point cane, and (3) quad cane. |
| Brodie et al., 2015 | SL | Triaxial accelerometers (Opal) | Auditory cues matched to each person's cadence. Five baseline walks no cues, then five conditions of five repeat walks cued at person's cadence and cued to various paired-step asymmetries of −10%, −5%, 0%, +5%, and +10%. |
| Fasano et al., 2016 | SL, SW, ST, SWT, joint ROM, double support time, speed | 3D Motion Capture (Qualisys), split-belt treadmill | Walking task: (1) Belts at same speed (tied) 5 min; (2) split w/worst side reduction (i.e., worst leg/shortest SL on slower belt), 10 min; (3) split w/best side reduction (i.e., best leg on slower belt), 10 min. |
| Kahn et al., 2009 | SL, stride length, ST, SWT | Pressure-Sensitive Mat (Gait Mat II), treadmill | Stepping task: Step w/impaired limb on treadmill while unimpaired held off treadmill, phase 1—single session unilateral stepping (UST), phase 2—repeated UST. UST for 20 min, 1 session in phase 1, 10 sessions in phase 2 over 2–3 weeks. |
| Lewek et al., 2012 | SL, step time | Split-belt treadmill | (1) 20 min treadmill walking followed by (2) 10–15 min overground training using the IVERT system to provide symmetry feedback in virtual reality. A total of 18 sessions over 6 weeks. |
| Lewek et al., 2018 | SL, step time | Pressure-sensitive mat (GAITRite), 3D motion capture (VICON), split-belt treadmill | Walking task: (1) 2 min belts tied, (2) 18 min training phase, (3) 10–15 min overground walking; 18 sessions; three groups: augmentation, minimization, or control. |
| Little et al., 2020 | SL, stride length, SLS (single limb support) %, gait speed | Pressure-sensitive mat (GAITRite), 3D motion capture (VICON), split-belt treadmill | Three treadmill conditions: Thera stride, 30% body weight support, and guidance of non-paretic limb. Maximum of 40 s for each condition. |
| Ma et al., 2018 | SL, SW, stance phase, single support phase | Pressure-sensitive mat (GAITRite) | Walked with and without textured insole. Two trials each condition. |
| Mieville et al., 2018 | ST, SL, trunk progression, forward foot placement | 3D motion capture (Optotrak Certus), split-belt treadmill | Walking task: (1) Baseline, tied belt at self-selected speed, 3 min; (2) perturbation, split belt w/slow belt at self-selected speed and fast belt at double the self-selected speed, 6 min; (3) post-perturbation, tied belt at self-selected speed, 3 min. Protocol completed twice with non-paretic (NP) on fast belt first. |
| Padmanabhan et al., 2020 | SL | 3D motion capture (VICON), split-belt treadmill | Three walking treadmill trials, each 4 min: (1) Without feedback (baseline), (2) participants preferred gait pattern with visual feedback, and (3) symmetry step lengths with visual feedback and instructed to hit same target. |
| Yen et al., 2015 | SL, SWT | Custom 3D position sensors | Ankle position signals triggered swing assistance or resistance to the affected leg via a cable-driven robot. Swing assistance and resistance sessions were 2 weeks apart. |
| Zanardi et al., 2019 | Max flexion (hip, knee, ankle), hip abduction ROM, ST, relative ST(%) | 3D motion capture (VICON), treadmill | Eleven weeks of Nordic Walking (NW): 4 sessions NW technique adaptation, 18 sessions NW training. |

SL = step length, SW = step width, ST = stance time, SWT = swing time.

The main training techniques for improving gait asymmetries were a split-belt treadmill ($n = 6$), followed by insoles/orthoses ($n = 3$). Other interventions utilized visual feedback ($n = 2$), canes ($n = 1$), auditory cues ($n = 1$), unilateral stepping task ($n = 1$), cable-driven robots ($n = 1$), and Nordic walking (i.e., using poles for arm participation to move the body forward; $n = 1$). The most common duration of the interventions was only one day ($n = 9$); other studies lasted from 2 to 11 weeks ($n = 5$).

### 3.4. Gait Symmetry Equations and Findings

Eight studies used a variety of symmetry ratio equations, as shown in Table 4. Five studies used the symmetry index (SI); however, two studies used a slightly different equation, while another study used a different method starting with a symmetry ratio to measure the symmetry index. One study used generalized estimating equations to quantify gait asymmetry. One study did not report the equation used.

**Table 4.** Summary of the Symmetry Equations and Findings of the included articles.

| References | Symmetry Equation | Findings |
|---|---|---|
| Afzal et al., 2015 | $R = \frac{\text{ST healthy side}}{\text{ST paretic side}}$ | Increased stance time and reduced gait asymmetry. |
| Arazapour et al., 2016 | $SI\% = \frac{XL - XR}{\frac{1}{2}(XL - XR)} \times 100$ | New powered KAFO decreases asymmetries in base width, swing time, stance phase %, and knee flexion during the swing phase. |
| Beauchamp et al., 2009 | $\text{Symmetry} = \frac{\frac{\text{paretic SWT}}{\text{paretic ST}}}{\frac{\text{nonparetic SWT}}{\text{nonparetic ST}}}$ | Single cane showed improvement in gait symmetry in subjects with baseline asymmetry. |
| Brodie et al., 2015 | $\text{PairedStepRatio}_n = \frac{\text{LeftStep}_n - \text{RightStep}_n}{\min(\text{LeftStep}_n \text{ or RightStep}_n)} \times 100$ | Auditory cues improved gait steadiness in most subjects with PD. Gait symmetry unaffected by symmetry matched auditory cues. |
| Fasano et al., 2016 | NR | Best reduction side led to worsening interlimb coordination, but improved spatial symmetry. Worst reduction side led to improved interlimb coordination, but decreased spatial symmetry. |
| Kahn et al., 2009 | $R = 100 - \left(\left\|\frac{\text{unimpaired SL}}{\text{impaired SL}}\right\| \times 100\right)$ | Phase 1: SLA improved by 9–13% and was maintained up to 24 h post-training, and ~12% improved single limb stance time of impaired limb. Phase 2: SLA decreased at 1 and 2 weeks post-training. |
| Lewek et al., 2012 | $SR = \frac{\text{paretic}}{\text{nonparetic}}$ | Improved step length and step time asymmetries. |
| Lewek et al., 2018 | $SR = \frac{\max(\text{paretic, nonparetic})}{(\text{paretic} + \text{nonparetic})}$ | All groups improved step length asymmetries from pre- to post-testing. No improvement in stance time with temporal training. |
| Little et al., 2020 | $\text{Temporal SI} = \frac{\text{SLS\%paretic}}{(\text{SLS\%paretic} - \text{SLS\%nonparetic})}$ | Guidance of non-paretic leg induced temporal symmetry by increasing paretic and decreasing nonparetic SLS% concurrently. Guidance of non-paretic leg induced spatial symmetry, but was not statistically significant. |
| Ma et al., 2018 | $SI = 2 \times \frac{(V_{\text{unaffected}} - V_{\text{affected}})}{(V_{\text{unaffected}} + V_{\text{affected}})} \times 100$ | Decreased stance phase and single support phase asymmetries |
| Mieville et al., 2018 | (1) $SR = \frac{\text{higher value}}{\text{lower value}}$<br>(2) normalized $SR = \frac{(SR - SR_{\text{group mean}})}{(SR_{\max} - SR_{\min})}$<br>(3) relative normalized $SR = \left[\frac{(1 - SR_{\min})}{(SR_{\max} - SR_{\min})}\right] + \text{normalized } SR_{\min}$<br>(4) normalized $SI = \left\|\frac{(\text{normalized } SR - \text{relative } SR_{\text{mean}})}{(\text{normalized } SR_{\max} - \text{normalized } SR_{\min})}\right\|$ | Reduced asymmetries in at least one spatiotemporal parameter in non-paretic fast and paretic fast conditions. |
| Padmanabhan et al., 2020 | $SLA = \frac{(SL_{\text{longer}} - SL_{\text{shorter}})}{(SL_{\text{longer}} + SL_{\text{shorter}})}$ | Improved step length asymmetries with symmetric stepping condition. |
| Yen et al., 2015 | $SI = \frac{X_a - X_s}{\frac{1}{2}(X_a + X_s)}$ | Improved step length symmetry after induced swing resistance in post-adaptation phase. |
| Zanardi et al., 2019 | Generalized Estimating Equation (GEE); $p < 0.05$ considered asymmetric | Improved degree of maximum knee flexion in less-affected limb and improved hip abduction range of motion. |

R = ratio, SR = symmetry ratio, SI = symmetry index, GAI = gait asymmetry index, SLA = step length asymmetry, SLS = single-limb support, ST = stance time, SWT = swing time, SL = step length, NR = Not Reported.

Almost all of the studies showed improvements in at least one spatiotemporal asymmetry (*n* = 13). Studies showed improved spatial asymmetries (*n* = 8), improved temporal asymmetries (*n* = 3), and spatial and temporal asymmetries (*n* = 3). The study that did not improve gait asymmetry did improve gait steadiness.

## 4. Discussion

The purpose of this systematic review was to investigate non-pharmacological interventions for improving gait asymmetries in neurological populations. Studies were included if they directly investigated interventions to improve gait asymmetry in adult clinical populations. This review included a total of 14 studies.

A significant finding of this review was the difference in the equations used to calculate gait asymmetries. This could potentially alter the results of the studies if the equations were consistent across all studies. Błażkiewicz and colleagues [42] compared four methods for calculating symmetry of gait: ratio index (RI), symmetry index (SI), gait asymmetry index (GA), and symmetry angle (SA) in healthy adults, and reported that the symmetry index was the most sensitive assessment of gait asymmetry even with potential artificial inflation. This is supported by an earlier study comparing the indices in identifying gait asymmetries in post-stroke participants [43]. Another common method for quantifying gait asymmetries is through statistical approaches (e.g., *t*-tests and cross-correlations) [44]. A few newer approaches have utilized similar indices as their foundation, but have either normalized or weighted the values to decrease the potential for artificial inflation [45,46]. The differences in equations and the different approaches should be considered for future studies examining gait asymmetries, and a consensus made on the most accurate method of detecting asymmetries.

Gait assessments and the gait variables measured were generally similar between most of the studies. A few of the studies included kinematic measures in addition to spatiotemporal measures. While kinetics (e.g., ground reaction forces) are commonly examined in amputee populations, assessment of gait kinetics was missing from this review on neurological populations. Kinetics are important as they relate to propulsion, weight bearing, and overall understanding of all aspects of gait asymmetries in clinical populations. The techniques for improving gait asymmetry varied widely between interventions, with the majority using split-belt treadmill training or the adoption of various insoles or orthoses. Future work could focus on implementing motor learning principles in their study design—as was suggested by Helm and Reisman [47]—to enhance gait rehabilitation. Several studies did note the researchers allowed the participants to use assistive devices, such as canes or ankle-foot orthoses, if used in normal walking. As a result, it is likely that the severity of the participants' gait impairments was greater than what was reported. Additionally, a few studies allowed participants to hold onto handrails, which has been shown to alter gait dynamics [48]. However, a balance should be struck between safety (need for handrail) and unconfounded study design (no handrail) when studying gait in clinical populations.

Overall, we observed many different methodologies for interventions to improve gait asymmetry, which was likely driven by heterogenous populations. Neurological populations have different pathological mechanisms that can affect gait. The part of the brain a stroke occurs in can vary and so can its effects. Overall, a stroke can cause acute and chronic motor impairments such as muscle weakness, spasticity, paralysis, and sensory impairments [49,50]. Parkinson's disease is frequently exhibited with dyskinesia, rigidity, bradykinesia, and increased gait impairments with dual task walking [3,51]. Poliomyelitis causes muscle weakness, fatigue, pain and muscular atrophy that lead to decreased gait ability [52,53]. Thus, interventions must be specifically designed with the target population in mind, as the mechanism of pathology is different between groups and even at the individual patient level. The majority of interventions were successful in improving gait asymmetry in at least one spatiotemporal or kinematic measure, with step length asymmetry being the most prominent measure. Spatiotemporal asymmetries are most

commonly measured and improved in gait asymmetry interventions, although Ryan and colleagues have suggested that spatiotemporal asymmetry measures may not be the best focus for improving gait function [54].

Several observations were made from these data that could help inform future research. First, approximately 70% of the populations included in this review were individuals with a stroke, showing a need for more research on other neurological populations (e.g., individuals with Parkinson's Disease, multiple sclerosis) or musculoskeletal issues (e.g., knee osteoarthritis or chronic ankle instability). Work in this area has been performed relative to other gait parameters [55–58], but not gait asymmetry. Second, the majority of the studies included in our review did not include a control group and contained sample sizes of 20 participants or less. However, the authors note the difficulty in recruiting and retaining participants from clinical populations, as often times inclusion and exclusion criteria need to be specific to exclude confounding factors that can influence gait such as secondary health problems. Additionally, due to the age of participants and the severity of their diseases, problems with retention can arise due to other health issues. Third, intervention durations were relatively short. Difficulties in recruitment and retention may explain why the duration of intervention for many of these studies was only 1 day long. Although, it should be noted that several studies lasted several weeks to almost three months, indicating that it is possible for longer duration interventions.

This review will allow for researchers and clinicians to explore methods for promoting the most successful intervention techniques to improve gait asymmetries. The kinetics of gait asymmetry were widely missing from the literature and are an important aspect to assess in gait rehabilitation. Future studies should also examine the frequency and duration of gait asymmetry interventions to determine the extent to which interventions may provide the greatest improvements, as well as create standard protocols based on disease pathology for improving gait asymmetries. Disease severity is associated with gait speed [59]. Increased gait speed, along with a more symmetric gait pattern, can reduce metabolic cost and decrease fall risk—thus improving aspects of quality of life measures [2–4,6,9]. It is important that future research also investigates the extent to which specific gait asymmetry metrics are associated with fall risk, as none of the studies included in this review assessed fall risk.

## 5. Conclusions

The purpose of this systematic review was to investigate non-pharmacological interventions for improving gait asymmetries. The literature suggests a wide range of methods may be used to improve spatiotemporal gait asymmetries, with split-belt treadmill training the most common method. Future research should further examine gait asymmetry kinematic and kinetic metrics, as these are greatly lacking in the literature. Additionally, the frequency and duration of various intervention strategies should be explored to achieve the greatest improvement in gait asymmetries. Additionally, it is important to continue to link specific gait asymmetry metrics to fall risk as well as to determine the best symmetry equation for quantifying gait asymmetries.

**Author Contributions:** Conceptualization, K.G.M. and C.K.R.; literature search, K.G.M. and C.T.L.; screening, K.G.M. and C.T.L.; data extraction, K.G.M. and C.T.L.; writing—original draft preparation, K.G.M.; writing—review and editing, K.G.M., C.T.L. and C.K.R. All authors have read and agreed to the published version of the manuscript.

**Funding:** This research received no external funding.

**Conflicts of Interest:** The authors declare no conflict of interest.

**Protocol Registration:** This review was not registered.

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
