# Peer review of "A Systematic Review of Non-Pharmacological Interventions to Improve Gait Asymmetries in Neurological Populations"

_symmetry, doi:10.3390/sym14020281_

Round 1

Reviewer 1 Report

In this systematic review, the authors review the literature on rehabilitation approaches for restoring spatiotemporal gait symmetry in persons with neurologic conditions (primarily stroke and Parkinson's disease with one study of poliomyelitis also included). The manuscript is well-written and the authors have motivated the need for this review nicely. The review is also straightforward and easy to read.

My primary suggestion for the authors is that the review seems much narrower in scope than its title suggests. First, this review covers primarily neurologic conditions and does not address orthopedic, surgical, or musculoskeletal conditions (e.g., sprains, fractures, ligament tears, amputations, osteoarthritis) that also cause gait asymmetry. Second, it focuses on spatiotemporal measures of gait asymmetry and does not cover other important gait parameters that are common targets of rehabilitation interventions that aim to improve gait symmetry (e.g., propulsion asymmetry after stroke). I suggest that the authors should either narrow the scope of the title, abstract, and introduction or provide a more comprehensive inclusion of these other relevant topics.

My other primary comment is that the discussion seems quite short. It would be helpful if the authors could synthesize their findings to a greater extent; what is helpful in the current literature, what is lacking, and what needs to be done next?

Reviewer 2 Report

Meder and colleagues provide a review of non-pharmacological interventions to specifically improve Gait Asymetries. The authors present the results of 14 articles. 
-To my mind the tittle has to precise this: “non-pharmacological” and has to take off the term “systematic”. It would be more precise if the title precise interventions to “specifically” improve gait asymmetries.
In the introduction, pharmacological treatment by Levodopa is evoked. It is not the subject of the paper. Many other pharmacological interventions are possible to improve gait in PD and so asymmetry even if asymmetry is not specifically studied. Many non pharmacological interventions may be evoked, deep brain stimulation, medullar stimulation,...
The populations are heterogeneous:
A remender of the difference between the gait particularity in PD (parkinson desease) or Stroke would be interesting.
Indeed, the asymmetry in PD is linked to the control of general motor symptoms, rigidity, bradykinésia, dyskinesia, double task difficulties...…
In stroke the mechanism of asymmetry may be different:  motor deficit, spasticity, heminegligence. Du to that, the intervention have to be be adapted to each condition. The author may evoke this clinical fact in their discussion.
In poliomyelitis this is still different.

 It would be interesting to precise the best criteria to define asymmetry of gait in each pathology.  
-no mention is made about the difference between  parameters evaluated in these different populations. Can we pool these studies? Please Justify
Maybe it would be more pertinent to focus on stroke, or PD. If not justify why. 
In the discussion I suggest that the author discuss the pertinence of the criteria and precise the result of Blazkiewicz.
Can the author precise why it is difficult to recruiting and retaining participants? Is that there is no quality of life improvement? 
COuld the author develop if the reduction of asymmetry is link to a reduction of fall or a better quality of life? Asymmetry is maybe an indicator of severity of the disease ( stroke or PD) and so a epiphenomenon to the gait alteration? If not please develop.

The manuscript should be reviewed carefully for grammatical and typographical errors

Introduction should be more concise and state clearly the aims of the study. 

The discussion could be strengthened

Reviewer 3 Report

In this Review paper the authors set forth to investigate non-pharmacological interventions to improve gait asymmetries. Although this paper is well written and is on an interesting area that is within the scope of the journal, there are several major concerns about the justification and the overall usefulness or contribution of the findings. The Introduction is not very clear about the reasons for doing the study and the gaps in the literature. At the discussion/conclusion section there is no clear message for the reader. Therefore, the usefulness of the paper is questionable.

Reviewer 4 Report

- In conclusions chapter, it would be good to detail alsso other aspects identified in the bibliographic research.
- Also, the calculation relations for the degree of asymmetry should be interpreted which of them is closest to the accuracy required for the determinations.
- The comparison of methods should also take into account the recording facilities used and the procedures applied.

Round 2

Reviewer 1 Report

The authors have addressed my prior comments.

Reviewer 2 Report

I appreciate the work of the authors. The authors have taken into consideration all the remarks of reviewers. 

Reviewer 3 Report

My recommendations have been fulfilled.